# *Streptococcus salivarius* as an Important Factor in Dental Biofilm Homeostasis: Influence on *Streptococcus mutans* and *Aggregatibacter actinomycetemcomitans* in Mixed Biofilm

**DOI:** 10.3390/ijms24087249

**Published:** 2023-04-14

**Authors:** Gabrijela Begić, Ivana Jelovica Badovinac, Ljerka Karleuša, Kristina Kralik, Olga Cvijanovic Peloza, Davor Kuiš, Ivana Gobin

**Affiliations:** 1Department of Microbiology and Parasitology, Faculty of Medicine, University of Rijeka, 51000 Rijeka, Croatia; gabrijela.begic@uniri.hr (G.B.);; 2Faculty of Physics and Centre for Micro- and Nanosciences and Technologies, University of Rijeka, 51000 Rijeka, Croatia; ijelov@phy.uniri.hr; 3Department of Physiology and Immunology, Faculty of Medicine, University of Rijeka, 51000 Rijeka, Croatia; ljerka.karleusa@uniri.hr; 4Department of Medical Statistics and Medical Informatics, Faculty of Medicine, Josip Juraj Strossmayer University of Osijek, 31000 Osijek, Croatia; kristina.kralik@gmail.com; 5Department of Anatomy, Faculty of Medicine, University of Rijeka, 51000 Rijeka, Croatia; olga.cvijanovic@uniri.hr; 6Department of Periodontology, Faculty of Dental Medicine, University of Rijeka, 51000 Rijeka, Croatia; 7Department of Dental Medicine, Faculty of Dental Medicine and Health, Josip Juraj Strossmayer University of Osijek, 31000 Osijek, Croatia; 8Clinical Hospital Centre, 51000 Rijeka, Croatia

**Keywords:** mixed oral biofilm, *A. actinomycetemcomitans*, *S. salivarius*, dental biofilm, dentin, hydroxyapatite, d-PTFE membranes

## Abstract

A disturbed balance within the dental biofilm can result in the dominance of cariogenic and periodontopathogenic species and disease development. Due to the failure of pharmacological treatment of biofilm infection, a preventive approach to promoting healthy oral microbiota is necessary. This study analyzed the influence of *Streptococcus salivarius* K12 on the development of a multispecies biofilm composed of *Streptococcus mutans*, *S. oralis* and *Aggregatibacter actinomycetemcomitans*. Four different materials were used: hydroxyapatite, dentin and two dense polytetrafluoroethylene (d-PTFE) membranes. Total bacteria, individual species and their proportions in the mixed biofilm were quantified. A qualitative analysis of the mixed biofilm was performed using scanning electron microscopy (SEM) and confocal laser scanning microscopy (CLSM). The results showed that in the presence of *S. salivarius* K 12 in the initial stage of biofilm development, the proportion of *S. mutans* was reduced, which resulted in the inhibition of microcolony development and the complex three-dimensional structure of the biofilm. In the mature biofilm, a significantly lower proportion of the periodontopathogenic species *A. actinomycetemcomitans* was found in the salivarius biofilm. Our results show that *S. salivarius* K 12 can inhibit the growth of pathogens in the dental biofilm and help maintain the physiological balance in the oral microbiome.

## 1. Introduction

The oral microbiome is a community consisting of about 500 commensal, symbiotic and pathogenic bacterial species [1,2]. A change in the balance of the oral microbiota leads to the development of oral diseases such as dental caries and periodontal and perioimplant diseases and conditions [3,4,5]. It is also associated with the development of systemic diseases, bacterial endocarditis, pneumonia and stroke [6,7,8,9,10].

Microorganisms from the oral cavity create a biofilm on all biotic and abiotic surfaces. The initial colonizers of the dental biofilm are streptococci, actinomycetes and veilonellae, which create attachment sites for other bacterial species [11,12]. The proportion of a particular species will depend on its ability to grow and outgrow neighboring cells. If there is a change in regulatory parameters such as nutrition, oral hygiene and host defense, favorable conditions can be created for pathogenic species within the biofilm and disease development [12,13].

As these diseases are not caused by a single pathogen but by biofilm microbial communities, classical pharmacological treatment has limitations. Bacteria in the biofilm are significantly more resistant to antimicrobial agents, and long-term use can cause suppression of healthy oral microbiota [14,15,16]. Numerous research studies have focused on inhibiting biofilm formation in terms of finding new materials with anti-adhesive properties [17,18,19]. The modern concept of combating biofilm infections is based on an ecological–bacterial approach, which aims to maintain physiological homeostasis within the biofilm. The measures of this preventive approach refer to promoting the growth of oral microbiota associated with health and reducing the virulent properties of biofilms [20].

Inspired by the promotion of healthy oral microbiota, this study aimed to examine whether *Streptococcus salivarius* K12 can influence the colonization and growth of other oral bacteria in a mixed biofilm. *S. salivarius* is a commensal bacterium, dominant in the oral cavity of healthy people. It has been proven that some strains have anti-inflammatory properties, produce bacteriocins and are antagonists to different bacterial species, including some with cariogenic potential [21,22,23]. Moreover, in recent clinical studies, it has been established that the administration of *S. salivarius* affects the improvement of clinical aspects in patients with COVID-19 [24,25].

In addition to *S. salivarius*, strains of *Streptococcus oralis*, *Streptococcus mutans* and *Aggregatibacter actinomycetemcomitans*—species commonly present in subgingival biofilms— were used in this study [26]. *S. mutans* is a species attributed with a highly cariogenic potential. This streptococcus is resistant to environmental stress and extremely resistant to low pH, which potentiates the breakdown of carbohydrates, thus stimulating other acidogenic species [27,28]. *S. oralis* is a member of the commensal oral microbiota, an opportunistic pathogen dominant in the initial stage of plaque formation, which can enhance the growth of potential pathogens in the plaque [29,30,31]. *A. actinomycetemcomitans* is a pathobiont that can play a crucial role in developing an infection by suppressing the host’s response. It is associated with periodontitis [9,32].

The study was conducted on four different materials: hydroxyapatite and dentin, which are most often used for in vitro studies of dental biofilm, and two d-PTFE membranes, used as barrier membranes in the alveolus preservation process. Results of this study could contribute to the development of an ecological–bacterial approach in the struggle against biofilm infections.

## 2. Results

### 2.1. Mixed Biofilm

#### 2.1.1. Total Number of Bacteria

In order to compare non-salivarius and salivarius mixed biofilms, the total number of bacteria was measured during biofilm formation. To see the difference, the number of bacteria in the stable biofilm 72 h after the addition of *A. actinomycetemcomitans* to the initial 24-h streptococcal biofilm is shown (Table 1). The total number of bacteria on d-PTFE membranes showed an increasing trend in the salivarius mixed biofilm, with a significant difference on the Permamem membrane. However, on dentin and hydroxyapatite, the number of bacteria in the salivarius mixed biofilm was significantly lower compared to the non-salivarius mixed biofilm.

#### 2.1.2. Number of Individual Bacterial Species

In order to investigate the difference between non-salivarius and salivarius mixed biofilms, the accompanying growth of certain bacterial species during biofilm formation was determined. The results are shown in Appendix A and Figure 1. The following observations can be drawn: (a) In the 24-h mixed salivarius streptococcal biofilm, *S. mutans* was represented in a significantly lower number than in the non-salivarius mixed biofilm. (b) In the 72-h mixed biofilm, the representation of *S. mutans* differed depending on the material on which the biofilm was formed. Lower values of CFU/mL were present in the salivarius mixed biofilm created on dentin and hydroxyapatite. In contrast, on d-PTFE membranes, lower values of CFU/mL were present in the mixed biofilm without *S. salivarius*. (c) In the 72-h biofilm, on all materials, the number of *A. actinomycetemcomitans* was significantly lower in the salivarius mixed biofilm.

#### 2.1.3. The Proportion of Individual Bacterial Species

To better understand the differences between non-salivarius and salivarius mixed biofilms, the proportion of individual species within the 24 and 72-h biofilms was calculated. The results are presented graphically. The initial streptococcal mixed biofilm created in the presence of *S. salivarius* K12 already showed a reduced proportion of *S. mutans* species after 24 h. The difference, although not equal, was observed in all materials (Figure 2). In the 72-h mixed biofilm, the most pronounced difference was observed in the proportion of *A. actinomycetemcomitans* in the biofilm with or without *S. salivarius*. The difference was determined on all materials. The proportion of non-salivarius biofilm ranged from 65% on Permamem d-PTFE membrane to 94% on dentin. A lower proportion of this bacterial species was recorded in the salivarius mixed biofilm, ranging from 5% on Permamem d-PTFE membrane to 63% on dentin (Figure 3).

#### 2.1.4. SEM Analysis of Mixed Biofilm

After examining the SEM micrographs, representative images were selected to show the difference in the structure of the 72-h mixed biofilm with and without the presence of *S. salivarius* K12 (Figure 4 and Figure 5). Figure 4a,b and Figure 5a,b show the salivarius mixed biofilm on Cytoplast d-PTFE and hydroxyapatite, respectively, structured as a disjointed organization of cells covered by a thin slime of Extracellular polymeric substances (EPS). In the non-salivarius mixed biofilm (Figure 4c,d and Figure 5c,d), a thicker biofilm layer was immediately observed. Biofilm has a complex three-dimensional structure consisting of microcolonies filled with EPS. EPS bridging between microcolonies (indicated by a red arrow) is also visible, leading to the formation of aggregates between microcolonies connected by channels (indicated by a yellow arrow) and the development of a complex biofilm. Similar results were obtained on the Permamem d-PTFE membrane and dentin.

#### 2.1.5. CLSM Analysis

##### Viability of Bacteria in Mixed Biofilm

Cell viability was assessed by examining the 72-h mixed biofilm after LIVE/DEAD staining. More bacterial biomass was observed in the non-salivarius mixed biofilm. However, biofilms were formed with predominantly living cells in the biofilm with and without *S. salivarius* K12 (Figure 6).

##### Cells and Biofilm Matrix in Mixed Biofilm

By staining cells and EPS within the 72-h mixed biofilm, we confirmed the results obtained by SEM analysis. The presence of *S. salivarius* K12 increased the biofilm’s biomass due to the creation of an EPS-rich matrix and the development of a complex biofilm structure (Figure 7). The average fluorescence intensity of the different experimental groups (for both the dead and live bacterial cells group and the EPS and live bacterial group) can be found in Figure 8.

## 3. Discussion

The literature points out that the most important step in controlling and preventing periodontal diseases is the inhibition of opportunistic pathogens in dental biofilm. The main role could be commensal bacteria, which antagonistically affect potential periodontal pathogens [4,33,34]. This study aimed to analyze whether the presence of *S. salivarius* K12 species in a mixed streptococci biofilm affects the colonization and growth of the periodontopathogenic species *A. actinomycetemcomitans*. Numerous studies confirm that this strain produces bacteriocins, salivaricin A2 and salivaricin B, which inhibit the growth of *S. pyogenes* [23,35,36]. In addition, it is an antagonist of bacterial species that cause halitosis [37,38,39]. The antagonistic effect of *S. salivarius* K12 on periodontopathogenic bacteria has been proven in a previous study (or studies). However, these tests were conducted in bacterial suspensions [40]. Since the action of pathogens causes periodontal diseases within a dysbiotically displaced biofilm, we wanted to examine the interactions within the biofilm in this study. Previously, the anti-adhesive and antibiofilm activity of another strain of *S. salivarius*, TOVE-R, was documented [41,42,43].

In this study, a mixed biofilm model with or without *S. salivarius* K12 was used to evaluate the antagonistic effect and to compare the number (CFU/mL) and the proportion (%) of individual bacterial species within the 72-h mixed biofilm. It was found that there was a difference that was observed in all tested materials. In the biofilm with the presence of *S. salivarius* K12, a different total number of bacteria was determined, depending on the material on which the biofilm was created. On dentin and hydroxyapatite, the total number of bacteria was lower in the presence of *S. salivarius*, while on d-PTFE membranes, the number increased (Table 1). It was also observed that in the 24-h salivarius–streptococcal mixed biofilm on the aforementioned membranes, there was a significantly higher proportion of *S. mutans* species (40–90%) compared to dentin and hydroxyapatite (10% and lower) (Figure 2). We believe that the physicochemical characteristics of the material’s surface and the surface of the bacterial cell itself influenced the degree of adhesion. D-PTFE membranes are hydrophobic polymers, dentin is a combined inorganic-organic material, and hydroxyapatite is a pure inorganic material. Due to their highly different chemical structures and compositions, they have different surface free energy (SFE) and surface roughness. The SFE surface is one of the main factors on which bacterial adhesion depends. Our previously published research determined the SFE of the tested bacterial species, and we proved that the difference between the SFE of bacteria and the surface they colonize affected adhesion [44,45,46,47,48]. Our results showed that in the presence of *S. salivarius*, in the initial 24-h streptococcal mixed biofilm, *S. mutans* was represented in smaller numbers and was a smaller proportion of the total number of bacteria. Events in this early phase of biofilm formation can influence population development over time and shift the balance from a non-virulent to a virulent degree.

*S. mutans* uses glucosyltransferase (Gtfs) to synthesize glucans from sucrose, contributing to bacterial attachment and subsequent colonization. The presence of Gtfs in the early stages of biofilm formation promotes the formation of a matrix rich in exopolysaccharides (EPS), which provides structural support for constructing three-dimensional structures of microcolonies in the biofilm [49,50,51]. The gtfB and gtfC genes that encode the production of Gtfs can be induced by autoinducers of other oral species in a mixed biofilm [52,53]. Furthermore, sucrose fermentation acidifies the matrix, favoring growth of acid-tolerant microorganisms [52]. The presence of 1% sucrose, as in the medium used in this study, increases biomass and the development of a three-dimensional structure mediated by *S. mutans*. Within these structures, niches with low pH favor the growth of acidogenic and aciduric pathogens protected by EPS [54]. This is consistent with our results. A thicker and more complex biofilm was observed in SEM micrographs showing non-salivarius mixed biofilm. The bacteria aggregated into microcolonies covered with EPS that filled the spaces between them, thus forming a 3D biofilm architecture. On the other hand, in the mixed biofilm with the presence of *S. salivarius*, the structural organization of accumulated cells covered by a thin layer of EPS was visible (Figure 4 and Figure 5). These results were confirmed by fluorescence microscopy. In the non-salivarius mixed biofilm, more bacterial biomass interwoven with a richer EPS matrix was observed (Figure 6 and Figure 7). *S. salivarius* shifted the balance by reducing the proportion of *S. mutans* already at the initial stage of biofilm development and inhibiting the formation of EPS-entangled microcolonies. In addition, weaker colonization of the periodontopathogenic species *A. actinomycetemcomitans* was observed in the salivarius mixed biofilm, resulting in a significantly lower number and proportion of these species in the 72-h biofilm (Table 1, Figure 3). It was previously confirmed that *S. salivarius* isolated from the oral cavity inhibits the growth of *A. actinomycetemcomitans* [55]. *A. actinomycetemcomitans* is a slow-growing species competing with fast-growing streptococci for the carbon substrate. This species prefers an alternative substrate, lactate, which allows it to survive in a competitive community [56,57]. However, it does not tolerate low pH levels. *S. salivarius* has a urease system and can lower pH by creating an alkaline microenvironment [58,59]. In addition, the EPS matrix created by *S. mutans* in non-salivarius mixed biofilm contains “pockets” within the microcolonies in which acids accumulate, resulting in spatially heterogeneous pH microenvironments and, ultimately, the heterogeneity of the microbial population within the biofilm [54,60]. Furthermore, *S. salivarius* produces bacteriocins that have an antagonistic effect on many, primarily Gram-positive bacteria [37,61,62]. The outer membrane of Gram-negative bacteria, such as *A. actinomycetemcomitans*, is impermeable to antibiotics [63]. However, the lactic acid in the biofilm can increase membrane permeability and sensitivity to antibiotics [64,65]. The results of this study show that *S. salivarius* K12 interacts with other species to have an antagonistic effect on the growth and dominance of opportunistic pathogens in dental biofilm.

## 4. Materials and Methods

### 4.1. Tested Materials

Materials independently evaluated in this study are d-PTFE membranes from different manufacturers, Permamem (Botiss biomaterials, Zossen, Germany) and Cytoplast (Osteogenics Biomedical, Lubbock, TX, USA), dentin (Immunodiagnostic Systems Holdings Ltd., Boldon, UK) and hydroxyapatite discs (Clarkson Chromatography Products Inc., South Williamsport, PA, USA). The membranes were aseptically cut to 5 × 5 mm, and t discs were 5 mm in diameter.

### 4.2. Bacterial Strains and Cultivation Conditions

Reference strains *Streptococcus mutans* ATCC 25175, *Streptococcus oralis* ATCC 6249, *Streptococcus salivarius* K12 ATCC BAA-1024, and *Aggregatibacter actinomycetemcomitans* ATCC 29522 (Microbiologics, St Cloud, MN, USA) were used. Bacteria were grown on blood agar plates (Biolife, Milan, Italy) supplemented with 5% sheep blood (Biognost, Zagreb, Croatia) in anaerobic conditions at 37 °C for 24–48 h.

### 4.3. Biofilm Development

An in vitro biofilm model with several bacterial species was developed. Pure bacterial cultures were grown to early stationary growth phase anaerobically in a modified protein-rich BHI liquid medium (Brain Heart Infusion, Becton, Dickinson and Company; Sparks, MD, USA) supplemented with 2.5 g/L mucin (Oxoid, Basingstoke, UK), 1.0 g/L yeast extract (Oxoid, Basingstoke, UK), 0.1 g/L cysteine (Sigma-Aldrich, Burlington, MA, USA), 2.0 g/L sodium bicarbonate (Merck, Darmstad, Germany), 5.0 mg/mL hemin (Sigma-Aldrich, Burlington, MA, USA), 1.0 mg/mL menadione (Merck, Darmstad, Germany) [26]. By measuring the optical density (OD 600), bacterial suspensions of selected streptococci with a concentration of 10^7^ CFU/mL were prepared and mixed in equal proportions. A suspension was prepared to form a mixed biofilm without the presence of *S. salivarius*—“non-salivarius mixed biofilm” with *Streptococcus mutans* and *S. oralis* and with the presence of *S. salivarius*—“salivarius mixed biofilm” (with *Streptococcus mutans*, *S. oralis* and *S. salivarius*).

Sterile materials, d-PTFE membranes, dentin discs and HA discs were placed in 96-well microtiter plate wells and conditioned for four hours at 30 °C, with 50% artificial saliva whose composition was described earlier [66].

Saliva was removed, and 200 µL of prepared mixed bacterial suspensions were added to all materials. The microtiter plates were incubated for 24 h at 37 °C under anaerobic conditions. After incubation, supernatant with planktonic bacteria was removed, and pure *A. actinomycetemcomitans* suspension was added to previously formed mixed streptococci biofilm on different materials. Microtiter plates with mixed biofilm formed on different materials were cultured for 72 h, with medium change every 24 h. The plates used to assess the sterility of the culture medium were used as controls.

### 4.4. Biofilm Quantification

In order to establish the development of individual species present in the mixed biofilm, the 24-h streptococcal biofilm (0 h) and the final mixed biofilm after 24 and 72 h were analyzed.

Bacteria in the mixed biofilm were detached by treatment in an ultrasonic bath (Bactosonic, Bandelin, Germany) at 40 kHz for 1 min. To quantify the bacteria, tenfold dilutions were plated on blood agar plates and CFU/mL was determined. For differentiation of individual species, dilutions were plated on Difco Mitis Salivarius agar with the addition of BBL Chapman Tellurite solution (Becton, Dickinson and Company; Sparks, MD, USA). All measurements were performed three times in triplicate.

### 4.5. Emission Scanning Electron Microscope (SEM) Analysis

The analysis of mixed bacterial biofilms on various materials was performed using a field emission scanning electron microscope—SEM (Jeol JSM-7800F), with a beam acceleration voltage of 7 kV and a working distance of 10 mm. Prior to SEM analyses, 72-h-old biofilms were washed in sterile PBS and air-dried in a sterile high-current chamber. They were then fixed with 4% glutaraldehyde and 0.5% paraformaldehyde (Sigma-Aldrich) prepared at 4 °C in 0.1 M phosphate buffer (Sigma-Aldrich) (pH 7.2) and then dehydrated by immersion in a series of increasing ethanol concentrations (50, 70, 80, 90, and 100%, Sigma-Aldrich). The samples under study were attached to a sample holder with conductive carbon tape. To prevent surface charging during electron beam irradiation, the samples were coated with a 5 nm thin layer of Au-Pd using the precision etching and coating system PECS II (Gatan Inc., Pleasanton, CA, USA).

### 4.6. Confocal Laser Scanning Microscope (CLSM) Analysis

#### 4.6.1. Testing the Viability of Bacteria in Biofilm

The viability of bacteria in the 72-h biofilm was tested using the LIVE/DEAD staining technique. LIVE/DEAD BacLight Bacterial Viability Kit L—7012 (Thermo Fisher Scientific, Waltham, MA, USA) was used in the staining procedure. Biofilms were grown as described in Section 4.3. on d-PTFE membranes. Biofilms were gently washed and covered with 5 µL of Propidium iodide and 5 µL of SYTO 9 in 990 µL of sterile distilled water. This was followed by incubation for 15 min at RT in the dark. After washing out the dye, an epifluorescence microscope with GFP/FITC (ex: 480 nm and em: 500 nm) and rhodamine (ex: 490 nm and em: 635 nm) filters were used for observation. All preparations were prepared in triplicate, and five visual fields of the prepared preparations were examined.

#### 4.6.2. Simultaneous Staining of Cells and Biofilm Matrix

FilmTracer SYPRO Ruby Biofilm Matrix Stain and SYTO 9 (Thermo Fisher Scientific, Waltham, MA, USA) were used to stain cells and biofilm matrix simultaneously. Biofilms were grown as described in Section 4.3. on d-PTFE membranes. Biofilms were gently washed and covered with 200 µL of SYPRO Ruby Biofilm Matrix Stain solution for matrix staining. The sample was incubated for 30 min at room temperature and protected from light. After gentle washing with sterile distilled water, bacterial cells were stained with diluted SYTO 9 dye for 3 min, followed by washing and observation using an epifluorescence microscope with GFP/FITC (ex: 480 nm and em: 500 nm) and rhodamine (ex: 580 nm and em: 700 nm) filters.

### 4.7. Statistical Analysis

The normality of the distribution of continuous variables was tested by the Shapiro-Wilk test. Continuous data were described by the median and 95% confidence interval for the median (95% CI). The Mann–Whitney U test was used to compare the median between two groups. All P values were two-sided. The level of significance was set at Alpha of 0.05. The statistical analysis was performed using MedCalc^®^ Statistical Software version 20.218 (MedCalc Software Ltd., Ostend, Belgium; https://www.medcalc.org (accessed on 1 April 2023).

## 5. Conclusions

The presence of *S. salivarius* K12 at the initial stage of streptococcal mixed biofilm formation inhibited the development of an EPS-rich matrix and three-dimensional biofilm structure mediated by *S. mutans.* There is less colonization by *A. actimomycetemcomitans* in the salivarius mixed streptococcal biofilm. It is present in significantly smaller numbers in its more mature phase than in the biofilm without *S. salivarius* K12.

Our results suggest that *S. salivarius* K 12 can maintain the biofilm’s physiological balance and reduce the growth of periodontal pathogens such as *A. actinomycetemcomitans*. In addition, our results could improve the preventive approach in the fight against biofilm infections.

## Figures and Tables

**Figure 1 ijms-24-07249-f001:**
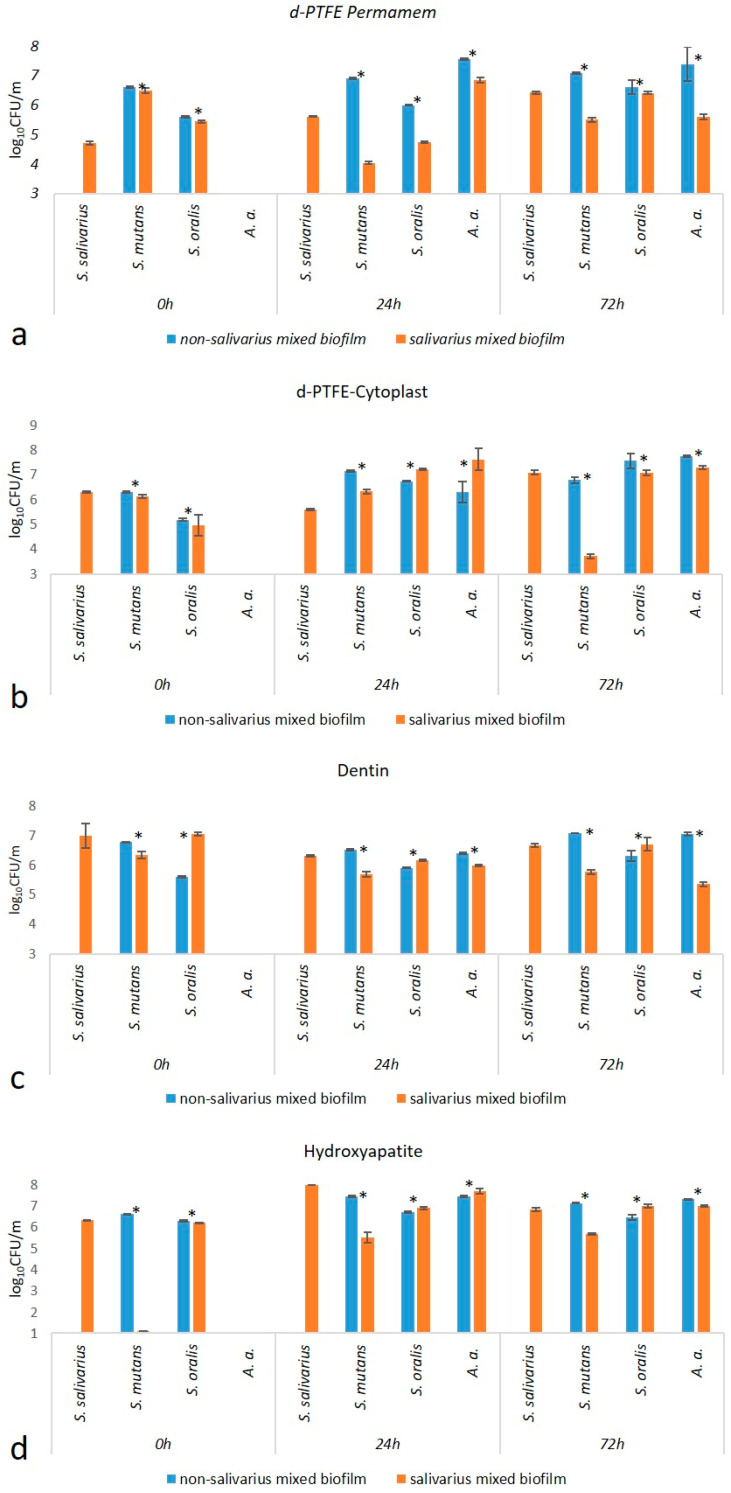
Comparison of the number (log_10_CFU/mL) of individual bacterial species in non-salivarius and salivarius mixed biofilms on: (**a**) Permamem, (**b**) Cytoplast, (**c**) Dentin and (**d**) Hydroxyapatite. 95% CI—95% Confidence interval: * Mann–Whitney U test.

**Figure 2 ijms-24-07249-f002:**
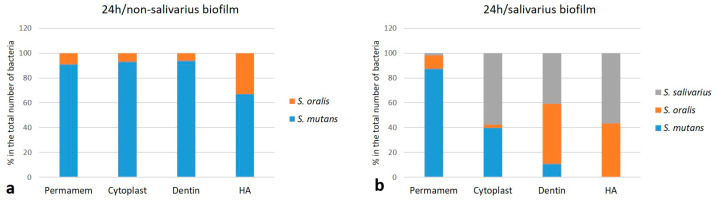
Distribution (%) of individual bacterial species in initial streptococcal biofilm (*n* = 9); (**a**) non-salivarius streptococcal mixed biofilm; (**b**) salivarius streptococcal mixed biofilm.

**Figure 3 ijms-24-07249-f003:**
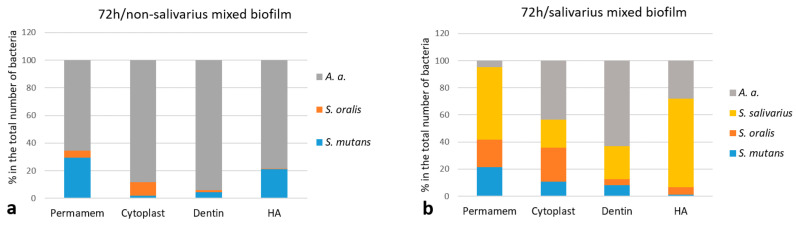
Distribution (%) of individual bacterial species on Permamem and Cytoplast d-PTFE membrane, dentin and hydroxyapatite (*n* = 9); (**a**) 72-h non-salivarius mixed biofilm; (**b**) salivarius mixed biofilm.

**Figure 4 ijms-24-07249-f004:**
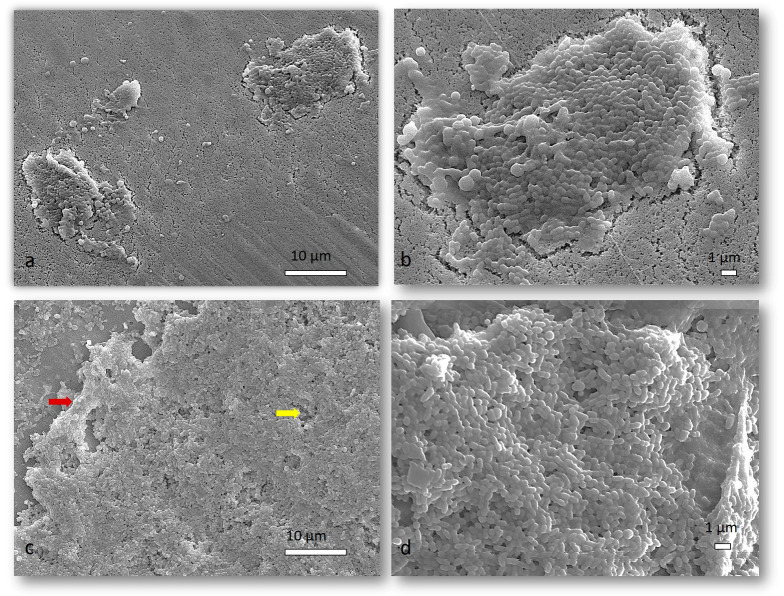
Representative SEM micrographs of a 72-h biofilm on Cytoplast d-PTFE. Different magnifications of salivarius mixed biofilm are presented in (**a**,**b**), while different magnifications of non-salivarius mixed biofilm are presented in (**c**,**d**); Red arrow—EPS bridging between microcolonies; Yellow arrow—aggregates between microcolonies connected by channels; Magnification of 2000× and 5000×.

**Figure 5 ijms-24-07249-f005:**
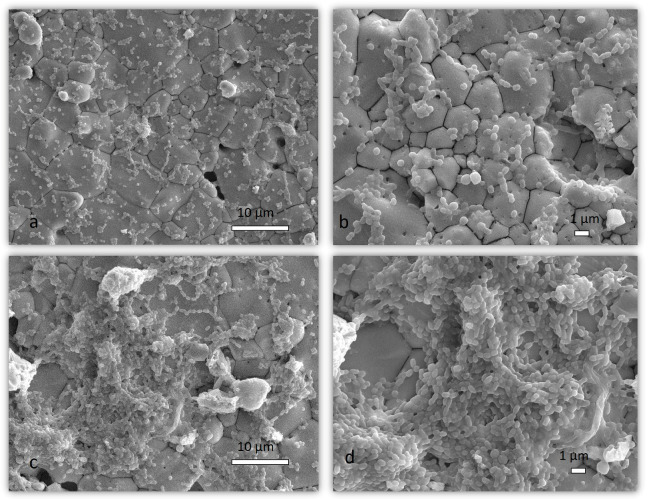
Representative SEM micrographs of a 72-h biofilm on hydroxyapatite. Different magnifications of salivarius mixed biofilm are presented in (**a**,**b**), while different magnifications of non-salivarius mixed biofilm are presented in (**c**,**d**); Magnification of 2000× and 5000×.

**Figure 6 ijms-24-07249-f006:**
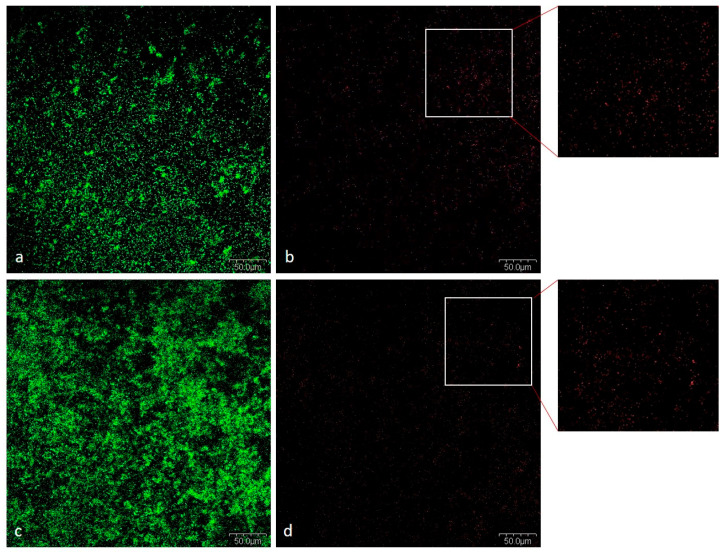
Illustration of dead (red) and live bacterial cells (green); (**a**) SYTO-9 stain in salivarius mixed biofilm; (**b**) Propidium iodide in salivarius mixed biofilm (**c**) SYTO-9 in non-salivarius mixed biofilm; (**d**) Propidium iodide in non-salivarius mixed biofilm; Magnification 20×.

**Figure 7 ijms-24-07249-f007:**
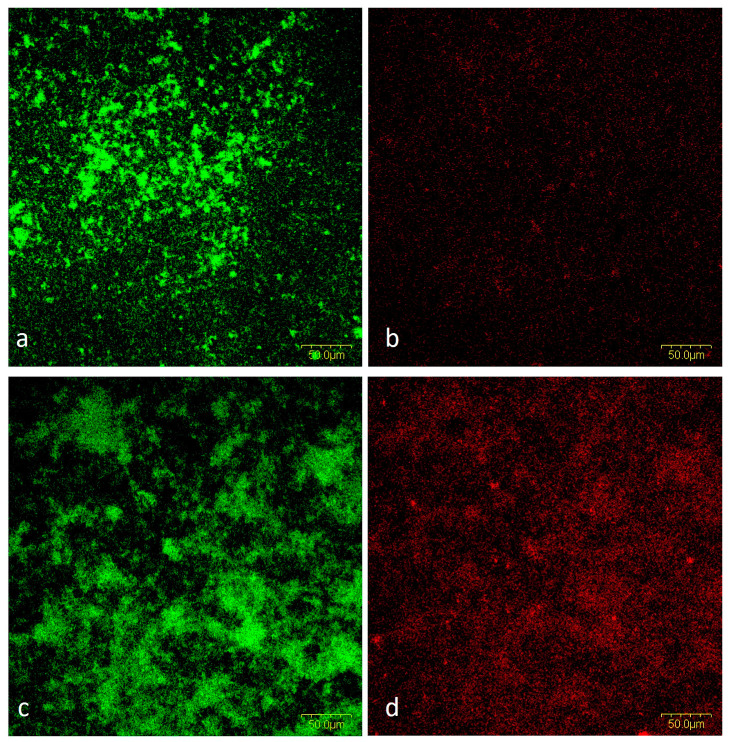
Illustration of the structural organization of EPS (red) and live bacterial cells (green); (**a**) SYTO-9 stain in salivarius mixed biofilm; (**b**) SYPRO Ruby Biofilm Matrix Stain in salivarius mixed biofilm; (**c**) SYTO-9 stain in non-salivarius mixed biofilm; (**d**) SYPRO Ruby Biofilm Matrix Stain in non-salivarius mixed biofilm; Magnification 20×.

**Figure 8 ijms-24-07249-f008:**
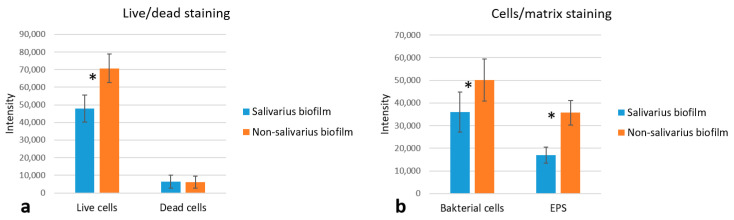
The average fluorescence intensity of the different experimental groups (*n* = 15). The asterisk (*) indicates statistically significant differences between the groups (*p* < 0.05); (**a**) Live/dead staining; (**b**) Cell/matrix staining.

**Table 1 ijms-24-07249-t001:** Comparison of the total number of bacteria (log CFU/mL) in a 72-h mixed biofilm.

	72 h		*p* *
	Non-Salivarius Mixed Biofilm	Salivarius Mixed Biofilm	
	Median (95% CI for Median)		
d-PTFE-Permamem	6.70 (6.12–6.94)	7.09 (6.95–7.15)	**0.005**
d-PTFE-Cytoplast	6.19 (5.73–7.13)	6.45 (6.29–6.53)	0.27
Dentin	6.89 (6.48–8.16)	5.75 (5.47–6.01)	**<0.001**
Hydroxyapatite	7.29 (5.39–7.83)	6.39 (5.95–6.77)	**0.04**

95% CI—95% Confidence interval: * Mann–Whitney U test; Bold denotes significant, (*n* = 9).

## Data Availability

Data sharing is not applicable to this article.

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
