# Peer review of "Streptococcus salivarius* as an Important Factor in Dental Biofilm Homeostasis: Influence on *Streptococcus mutans* and *Aggregatibacter actinomycetemcomitans* in Mixed Biofilm"

_ijms, 2023, doi:10.3390/ijms24087249_

Round 1

Reviewer 1 Report

This manuscript explored the antibiofilm effect of Streptococcus salivarius in the mixed culture of oral bacteria on different dental materials' surfaces. The author found S. salivarius could repress the growth of Streptococcus mutans and Aggregatibacter actinomycetemcomitans in 24h and 72h biofilm, respectively. The SEM and confocal microscopy images also confirmed that S. salivarius could reduce the EPS in dental biofilm. The study introduces a new possible approach to control dental plaque. I think several critical issues need to be addressed in the manuscript before being published.

Major comments:

1. The manuscript tested 4 different dental materials, and S. salivarius only have a reduction effect on total bacteria & S. mutans numbers on dentin and hydroxyapatite surfaces, while two d-PTFE surfaces had contrasting results. The author only generally stated this may be due to the difference in the physicochemical property of surfaces. I think the authors need to provide more detailed information (experiment's results, hypothesis, or references) to explain this.

2. For SEM and CLSM, the manuscript only presented the results of one d-PTFE surface (which one?). Have authors also imaged biofilm on the other 3 materials? Did they have similar results as d-PTFE?

3. How many replicated samples did the authors test in each table and Figures 1, 2, and 6? Also, figure 6 needs p values and S.D.

4. The labels (1a,1b, 2a,2b...) of figure 3 are quite confusing, I suggest using Figure 3a, 3b, 3c, and 3d as labels. Same suggestion for Figure 4 and 5.

5. I suggest removing merged images in Figure 4 and 5. Also, the PI-stained images look very dim in figure 4, does it mean the number of dead cells is very low?

Minor comments:

1. Please make sure to use the correct/unified name of materials, such as "hydroxyapatite" not "hydroksyapatite".

2. For table 2, it's better to use a bar figure to show these results.

3. I suggest using the unified format for figure 1 and 2.

4. For figure 3, do the authors have the SEM image of the bare surface without biofilm? 

Reviewer 2 Report

Begić et. al., have done a very nice piece of work. I will suggest few things,

1. Line 285 in vitro should be in italic.

2. It is good that you summarized the non-salivarius and salivarius mixed biofilms with individual species at different time points (24 and 72-hour). I suggest representing the Table 2 in different way to make it more convenient for readers to understand.

3. In the Figure 4 (viability assay) dead cells are not clearly visible to differentiate the live cells. It is very hard to see the dead cells in the images. Propidium iodide (red) is not expressed clearly. I believe authors need to take clear images to address this issue.

Reviewer 3 Report

The manuscript of “Streptococcus salivarius as important factor in dental biofilm 2 homeostasis: influence on Streptococcus mutans and 3 Aggregatibacter actinomycetemcomitans in mixed biofilmis interesting to read and appreciated for good attempt. Authors explained about several things as these 83 studies can contribute to development of a bacterial-ecological approach in the struggle 84 against biofilm infections. I decided minor revision and also correct the following suggestion before accept the manuscript.

1.     The abstract should be written corrected with obtained results and limited words.

2.      Hypothesis of introduction part is very limited. Need current explanation.

3.      Some many references are old like ten years ago, try to add new references.

4.      There are some typo and grammatical error present in the manuscript

5.     Very basic experimental studies were conducted on this manuscript and results don’t have any statistical analysis and there is no any novelty

6.       Figures legends are not reasonable 

Round 2

Reviewer 1 Report

All my comments have been addressed, I have no more question for the manuscript.